# Recent Research and Application Prospect of Functional Oligosaccharides on Intestinal Disease Treatment

**DOI:** 10.3390/molecules27217622

**Published:** 2022-11-07

**Authors:** Tong Xu, Ruijie Sun, Yuchen Zhang, Chen Zhang, Yujing Wang, Zhuo A. Wang, Yuguang Du

**Affiliations:** 1State Key Laboratory of Biochemical Engineering, Institute of Process Engineering, Chinese Academy of Sciences, Beijing 100190, China; 2School of Chemical Engineering, University of Chinese Academy of Sciences, Beijing 100049, China; 3Key Laboratory of Carbohydrate Chemistry and Biotechnology, Ministry of Education, School of Biotechnology, Jiangnan University, Wuxi 214000, China

**Keywords:** functional oligosaccharides, gut microbiota, intestinal barriers, intestinal diseases

## Abstract

The intestinal tract is an essential digestive organ of the human body, and damage to the intestinal barrier will lead to various diseases. Functional oligosaccharides are carbohydrates with a low degree of polymerization and exhibit beneficial effects on human intestinal health. Laboratory experiments and clinical studies indicate that functional oligosaccharides repair the damaged intestinal tract and maintain intestinal homeostasis by regulating intestinal barrier function, immune response, and intestinal microbial composition. Functional oligosaccharides treat intestinal disease such as inflammatory bowel disease (IBD) and colorectal cancer (CRC) and have excellent prospects for therapeutic application. Here, we present an overview of the recent research into the effects of functional oligosaccharides on intestinal health.

## 1. Introduction

With the continuous improvement of people’s quality of life in recent years, the diet structure has changed from simple to complex. Today, the global nutrition situation is complicated. On the one hand, hunger and malnutrition are the dominant concerns in low- and middle-income countries. On the other hand, millions of people are at increased risk of developing diet-related chronic diseases, for example, intestinal disease, heart disease, and diabetes. Taking inflammatory bowel disease (IBD) as an example, the incidence of IBD has increased year by year worldwide over the past decade [1,2], with the highest incidence of IBD in developed countries [3]. More than 2 million people in North America and 3.2 million people in Europe are afflicted with IBD [4]. With the development trend of globalization, IBD is becoming more and more common in developing countries such as Brazil and China [5]. In Brazil, Crohn’s disease (CD) and ulcerative colitis (UC) increased by 11.1% and 14.9%, respectively, from 1988 to 2012 [4]. In China, there were 350,000 IBD patients in 2014, which is expected to increase by 4.2 times by 2025 and an approximate 70% increase in UC and 30% in CD (data from CCDC). IBD is also a risk factor for colitis-associated colorectal cancer (CA-CRC), which causes death in about 15% of patients with IBD [6]. In 2020, the global number of CRC cases was close to 2,000,000 and accounted for 9.7% of the global cancer population (data from IARC). There are 1.5 million CRC patients in the United States. In recent years, the incidence and mortality of CRC have decreased, but there are still about 150,000 new patients each year [7]. In contrast, the number of CRC cases in China has been progressively higher than in the United States in recent years, with 2.6 times the patients of the United States. Chinese CRC patients account for 31% of the patients worldwide. From 1990 to 2019, the number of CRC cases in China increased by 700% (data from World Bank IHME-GBD). In addition, irritable bowel syndrome (IBS) is one of the most common intestinal disorders in clinical practice. The prevalence of IBS in Western countries is 10% to 20%, and the prevalence of IBS in China is 5.7%. In 2016, there were approximately 754 million people with IBS worldwide, and it is expected to reach 830 million by 2025 (data from Data monitor). In summary, with the increasing number of intestinal diseases represented by IBD, IBS, and CRC, their prevention and treatment is gradually becoming an important issue for domestic and international research.

The intestine is a vital digestive organ responsible for the digestion and absorption of nutrients, and the intestinal barrier prevents pathogenic bacteria, toxins, and other harmful substances from entering the intestine’s circulatory system [8,9,10]. The intestinal barrier is comprised of the epithelial and mucus barrier, immune barrier, and biological barrier, which together maintain the health and homeostasis of the intestinal tract. The intestinal epithelial and mucus barrier is mainly composed of single-layer cells connecting proteins and chemical substances in the intestinal epithelium. A variety of transmembrane proteins further constitute a complex protein network between adjacent cells. The integrity of the intestinal epithelial barrier depends on the link complexes in the protein network, including the tight junction, adhesion junction, and bridge and gap junction [11]. The chemical substances are composed of mucus, digestive fluid, antibacterial components, and other compounds secreted by the intestinal mucosa and microorganisms. The epithelial and mucus barrier prevents the penetration of harmful bacteria and toxins [12]. The intestinal tract is also the largest immune organ in the human body. The intestinal immune barrier includes intestinal-related lymphoid tissue (GALT), diffuse immune cells, and immune factors [13]. Microorganisms colonized in the intestine are considered intestinal biological barriers. Many laboratory and clinical studies have confirmed that the damaged intestinal barrier may lead to overactive immune responses in the intestinal microenvironment or the uncontrolled growth of microbial flora, leading to various diseases [14]. The effect of functional oligosaccharides on intestinal barrier function and health is illustrated in Figure 1.

Current treatment strategies for intestinal diseases include micro-ecological regulation therapy [15], surgical treatment [16,17], and drug therapy [17,18]. In recent years, micro-ecological agents including probiotics, prebiotics, and diet fibers have drawn more and more attention to treating intestinal diseases. The International Association for Probiotics and Prebiotics (ISAPP) redefined prebiotics in 2016: the host microorganisms selectively use them to make them healthy substrates [19,20]. Prebiotics are diverse and are divided into carbohydrate sources and non-carbohydrate sources, with functional oligosaccharides as the principal source. Functional oligosaccharides are carbohydrate oligomers with branched or straight chains of 2–20 monosaccharide molecules linked through glycosidic bonds. Here we mainly introduce the representative functional oligosaccharides: isomaltooligosaccharide (IMO), fructooligosaccharides (FOS), xylooligosaccharides (XOS), galactooligosaccharides (GOS), chitosan oligosaccharides (COS), and human milk oligosaccharides (HMOs). The structure of the functional oligosaccharides is shown in Figure 2.

Various studies have shown that functional oligosaccharides can ease intestinal injury and treat intestinal diseases by maintaining and repairing intestinal barriers [21,22,23]. As of April 2022, there were 112 registered clinical trials (data from ClinicalTrials.gov, accessed on 1 June 2022) related to assessing beneficial effects of functional oligosaccharides on human health, including 28 IBS-related studies and 33 IBD-related studies (Some details of the studies are shown in Table 1).

It is well accepted that functional oligosaccharides such as raffinose oligosaccharide (ROS) [24], FOS, and GOS [25] can affect specific groups of the microbial community in vitro and in vivo to promote their growth and metabolic activity, thereby maintaining host gut health benefits [26]. In addition, functional oligosaccharides are also considered to interact directly with the host and exert local positive effects on inflammation and barrier function by regulating immunity and intestinal epithelial cell signal transduction [27]. Functional oligosaccharides have different effects on the host intestine due to their different monosaccharide composition, degree of polymerization, and linkage types [28,29]. A number of studies have been carried out regarding the activity of functional oligosaccharides affecting intestinal barrier function. This review focuses on the latest research on functional oligosaccharides and their effects on intestinal health, especially their interaction with intestinal flora, immunity, and disease treatment.

## 2. Effects of Functional Oligosaccharides on the Intestinal Barrier

### 2.1. Biological Barriers

Functional oligosaccharides can be selectively fermented into short-chain fatty acids (SCFA) in the gut [30] to maintain intestinal function and health by controlling the growth of pathogenic microorganisms, reducing pH, preventing peptide degradation, and the formation of toxic compounds [31,32]. Functional oligosaccharides can be used directly by the microbiota as a carbon source. Furthermore, some studies have found that functional oligosaccharides such as inulin-derived FOS can also increase the colonization sites of probiotics in the intestinal tract [33]. Our previous studies also found that COS promoted the growth, metabolic activity, and metabolite concentration changes of probiotics represented by *Akkermansia muciniphila* by affecting specific populations in microbial groups; reduced the adhesion, invasion, and colonization of intestinal pathogens represented by *Escherichia coli*; and inhibited the occurrence and development of enteritis, thereby maintaining intestinal health [34,35].

There is a correspondence between functional oligosaccharides and probiotics. Functional oligosaccharides exhibit a complex degree of polymerization and glycosidic bonding [36,37], and probiotics utilize functional oligosaccharides with a diversity of transporter proteins and glycosidic hydrolases [38,39]. Therefore, the growth promotion effects of functional oligosaccharides on probiotics are species-specific. For example, butyrate-producing strains showed different growth curves in the presence of FOS, GOS, and XOS [40,41]. The same kind of FOS, due to their different sources, have different effects on the growth of probiotics. Studies have shown that inulin FOS have more noticeable effects on the growth of *Bifidobacterium* than sucrose FOS. The molecular mechanism of the metabolism of FOS, GOS, and milk-derived oligosaccharides by probiotics has also been studied, and the unique intake mechanism for functional oligosaccharides plays an active role. A brief summary of the coincidence relationship between common functional oligosaccharides and probiotics is shown in Table 2.

Based on previous and our own research on functional oligosaccharides affecting proliferation and colonization of probiotics and considering the specificity and complexity of the interaction between intestinal flora and functional oligosaccharides, it is critical to further study the effects of functional oligosaccharides with different structures on the changes in intestinal metabolites, bacterial gene expression, and potential molecular mechanisms in maintaining intestinal barrier function.

### 2.2. Immune Barrier

Functional oligosaccharide plays a positive role in the intestinal immune barrier. Indirectly, functional oligosaccharides can be fermented by probiotic to produce SCFA, which regulate the activity of T cells, B cells, and dendritic cells [14,64]. For example, oral administration of FOS increased the level of SCFA, including butyrate, which increased the level of regulatory T cells in the mesenteric lymph nodes of mice [65,66]. In addition, some functional oligosaccharides have also been found to directly act on intestinal-associated immune cells and immune factors, providing beneficial effects on intestinal diseases such as allergies or IBD [67,68]. Specifically, functional oligosaccharides can stimulate Toll-like receptors and induce the differentiation of immune cells represented by T and B cells to regulate intestinal immunity [67,68]. Functional oligosaccharides also regulate the secretion of inflammatory factors represented by IFN-γ, IL-5, and IL-6 in the intestine and increase the content of immunoglobulin represented by IgA, IgM, and IgG. For example, some studies have found that FOS and GOS act as TLR4 agonists in intestinal epithelial cells; activating the TLR4-NF-κB pathway; and reducing pro-inflammatory factors such as IL-12p35, IL-8, and TNF-α [69]. FOS and arabinogalactan oligosaccharides regulate the immune-related parameters in GALT, secondary lymphoid tissue, and peripheral circulation [70]. We summarize the regulatory effects of different functional oligosaccharides on the intestinal immune barrier in Figure 3 and Table 3.

### 2.3. Epithelial and Mucus Barrier

The human gastrointestinal tract has no relevant enzyme system to hydrolyze functional oligosaccharides [90,91]. However, functional oligosaccharides exhibit excellent benefits in the composition and maintenance of intestinal epithelium, either directly or indirectly. It is well accepted that functional oligosaccharides are utilized by gut microbes [92,93] to produce metabolites such as short-chain fatty acids (SCFA), which regulate host cell growth, differentiation, apoptosis, and physiological functions in the intestine [94,95]. Moreover, recent evidence suggests that functional oligosaccharides such as COS, GOS, and cello-oligosaccharides could directly affect the permeability and integrity of intestinal epithelial cells by improving colonic epithelial cell transmembrane resistance and reducing intestinal epithelial cell permeability to fluorescein isothiocyanate-glucan [96,97]. Studies show that functional oligosaccharides could upregulate the expression of specific tight junction proteins of epithelial cells [98]. The mechanism of functional oligosaccharides regulating intestinal epithelial cell homeostasis has not been fully explored.

Functional oligosaccharides can also affect the production of mucin and antimicrobial peptides by host cells [97,99]. For example, feeding 1 g/d of GOS to rats with severe pancreatitis can significantly improve their mucus defects [100]. This improvement effect is related to the structure of functional oligosaccharides and the dose of the functional oligosaccharides used. However, only a few studies considered exploring the protective effect of a functional oligosaccharide dose on intestinal mucus barrier function. The study noted that berberine promoted the proliferation of *Akkermansia* in a dose- and time-dependent manner in mice, with 300 mg/kg of berberine showing a two-fold higher proliferation rate than 200 mg/kg. The investigators also demonstrated that this proliferation works by promoting the secretion of mucins, especially mucin-2 [101]. The structure-function relationship and action mechanism of functional oligosaccharides need to be further analyzed and evaluated.

## 3. Application of Functional Oligosaccharides in Intestinal Diseases

### 3.1. Colorectal Cancer

The colon environment, including imbalanced intestinal microflora and mutations in the Wnt signaling pathway are the leading causes of CRC [102,103]. The current treatments for CRC include chemotherapy, radiotherapy, and surgery, but most of them are accompanied by high-risk complications, and the success rate is limited. Therefore, new early treatment strategies are needed [104]. The use of functional oligosaccharides in preventing CRC may be promising. Studies show FOS and GOS can reduce the severity of colon cancer in rats and mice induced by 1,2-dimethylhydrazine by reducing the number of colon ACF [105,106,107]. Researchers have found that low-degree FOS are more effective in treating early colon cancer in mice induced by DMH [108] and significantly reducing the risk of colon cancer in animal models [109]. There are two aspects regarding the inhibitory effect of functional oligosaccharides on colorectal cancer. First, functional oligosaccharides affect the homeostasis of intestinal microflora by promoting the growth and colonization of intestinal probiotics and upregulating production of metabolites such as SCFA, which inhibit the proliferation and differentiation of colon tumor cells [104,110] and regulates exogenous metabolic enzymes that stimulate the activation and metabolism of carcinogens [111,112]. Furthermore, functional oligosaccharides directly regulate the functions of intestinal GALT and other immune cells, influence gene expression levels of cancer cells, and promote cancer cell apoptosis [109].

The clinical data also show that functional oligosaccharides have a positive effect on the immunological indexes of colon cancer and microbial flora abundance [113]. However, some clinical data point out that functional oligosaccharides do not significantly reduce the mortality of colorectal cancer in women after menopause [102,113]. There is no clear explanation for the structure-activity relationship, dosage, and individual differences of functional oligosaccharides, which may also be the main reason for restricting the clinical trials of functional oligosaccharides in the treatment of colorectal cancer. Therefore, the clinical treatment of CRC with functional oligosaccharides remains unconfirmed. Consequently, research on new technologies such as combining probiotics and functional oligosaccharides as targeted therapeutic agents for colon cancer based on host–guest chemistry is also an aspect worth exploring [114].

### 3.2. Inflammatory Bowel Disease

IBD is a chronic nonspecific gastrointestinal inflammatory disease that destroys the intestinal mucosal structure and floral balance, leading to abnormal systemic biochemical indexes [115]. The etiology of inflammatory bowel disease is not clear, while comprehensive factors such as intestinal flora, immunity, environment, and gene susceptibility might be involved.

The DSS-induced mouse colitis model is one of the widely recognized models for studying the pathogenesis of IBD and evaluating potential therapeutic methods [116]. Growing evidence supports the potential of functional oligosaccharides to treat inflammatory diseases, including colitis. FOS and GOS in vitro affect immunity by binding to TLR on monocytes, macrophages, and intestinal epithelial cells and regulating cytokine production and immune cell maturation [69,117,118,119,120]. In addition, animal models and clinical studies have shown that functional oligosaccharides reduce the intestinal inflammatory response and IBD symptoms [121,122]. A clinical study focused on enteritis after abdominal radiotherapy (RT) found that FOS supplementation in patients’ daily diet can stimulate the proliferation of *Lactobacilus* and *Bifidobacterium*, thereby repairing intestinal mucosal damage during RT and preventing the occurrence and development of IBD [123]. Our previous studies have found that COS treatment upregulates the expression of occludin in the proximal colon of diabetic mice [34], alleviates DSS-induced mucosal defects in IBD, and protects the intestinal mucosal barrier function of ulcerative colitis mice [97].

Future studies need to understand how functional oligosaccharides regulate the disease-related signaling pathways, drive different cellular processes and regulate intestinal functions, and conduct the mechanism of functional oligosaccharides as a drug adjuvant or substitute in the treatment of IBD.

### 3.3. Irritable Bowel Syndrome

IBS is a chronic disease affected by stress and eating habits. It is characterized by abdominal pain, mucosal and immune functions, and changes in the intestinal microbial structure. Dietary patterns, the intestinal microbial structure, inflammatory response, and other factors can aggravate the symptoms of irritable bowel syndrome. Dietary interventions are recommended to control the disease due to the efficacy and tolerance of common drug treatments.

Evidence shows that the ecological imbalance of intestinal and mucosal colon microflora in IBS is usually characterized by the reduction of the *Bifidobacterium* species [124,125,126,127]. Some studies have found that supplementing probiotics to regulate intestinal microflora are effective in treating IBS [128,129]. Some clinical studies have also found that low-dose functional oligosaccharides, such as FOS, can alleviate the symptoms of IBS patients through increasing the concentration of SCFA [130]. In contrast, a low FODMAP diet has gradually become the standard method for the treatment of IBS worldwide. This method can alleviate the clinical symptoms of IBS patients by limiting the daily intake of short-chain fermentable carbohydrates (low fermentable oligosaccharides, disaccharides, monosaccharides, and polyols (FODMAP)). Studies have consistently proven the clinical efficacy of a low FODMAP diet in patients with IBS [130]. In fact, the low FODMAP diet has clinical efficacy, but it reduces the abundance of intestinal *Bifidobacterium*, which is not conducive to the thorough treatment of IBS patients. In view of the pathogenic factors and pathogenesis of IBS and the complexity and diversity of individual microbial communities, we need to consider these two interventions for further research and consider individualized diagnoses according to clinical symptoms.

## 4. Application Prospect of Functional Oligosaccharides in the Intestinal Tract

Glycans generally have complex monosaccharide composition, glycosidic bond type and degree of polymerization, and their structural complexity is much higher than that of proteins and nucleic acids. In the past decade, glycoscience, with the support of governments, has made a lot of progress, and it has revealed the role of glycans in inflammatory responses and immune system regulation, cardiovascular diseases, intestinal diseases, and cancers. A variety of functional oligosaccharides have shown kinds of activities in intestinal barrier protection and repair and have demonstrated the great promise of glycans in intestinal disease treatment. However, the structure-activity relationship and molecular mechanism have not been fully elucidated. Glycan-based products used in related research are often a mixture of glycans with slightly different structural characters and subject to variations in different preparation methods and raw material sources. Recent studies have shown that small changes in the structure of glycans have significant effects on their activities, so the accurate analysis and preparation of glycan products is the key to clarify their structure-activity relationships and develop functional glycan products, for example, structural analysis and pharmacokinetic study of glycans by liquid chromatography-tandem mass spectrometry (LC-MS/MS) [131]. On the other hand, based on the different activities of different sugar chains, the combination of several different glycans have received more attention and applications [64,132,133]. In the face of the complex microbial and host environment in the intestinal tract, products containing multiple different glycans will play a greater role. However, there is a lack of in-depth research on the compounding mechanism and synergistic effect of multiple glycan recipes.

## Figures and Tables

**Figure 1 molecules-27-07622-f001:**
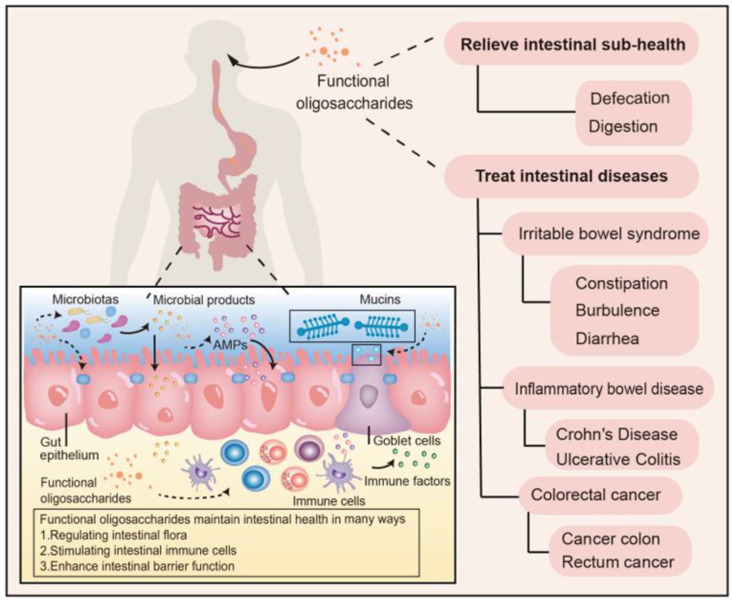
Schematic overview of the effects of functional oligosaccharides on intestinal barrier function and health. AMPs: antimicrobial peptides.

**Figure 2 molecules-27-07622-f002:**
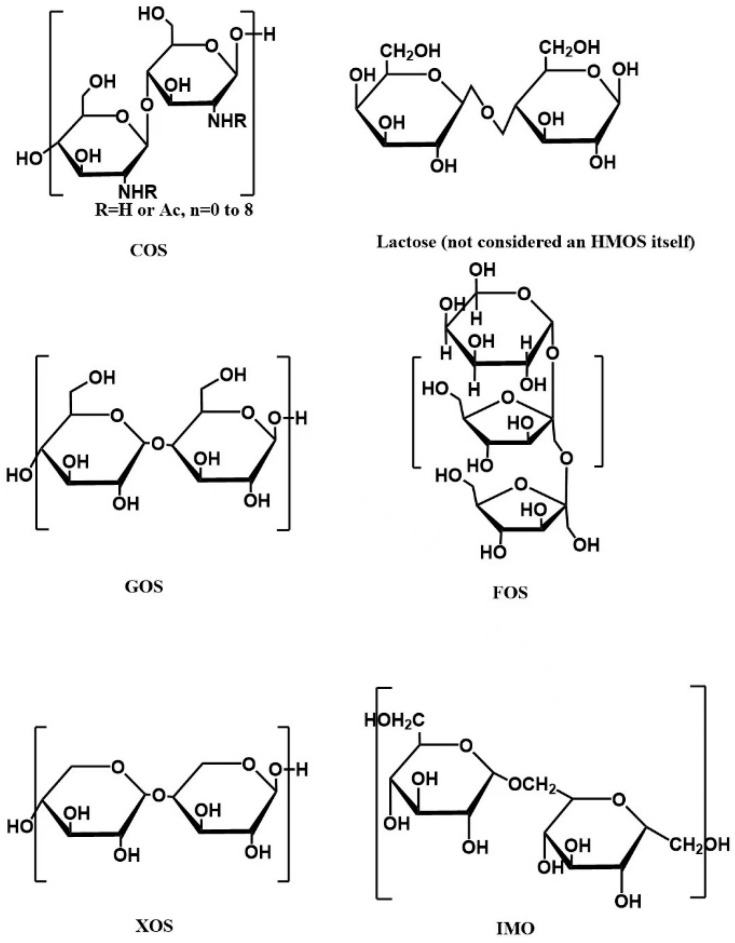
Schematic diagram of the structure of common functional oligosaccharides.

**Figure 3 molecules-27-07622-f003:**
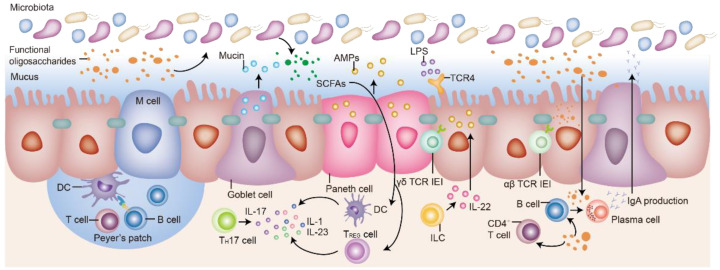
Schematic diagram of the modulating effect of functional oligosaccharides on the intestinal immune barrier. SCFAs: short chain fatty acids; AMPs: antimicrobial peptides; LPS: lipopolysaccharide; M cell: membranous/microfold cell; DC: dendritic cell; T cell: T-lymphocyte cell; B cell: B-lymphocyte cell; Th17 cell: T helper cell 17; IL-1/17/22/23: interleukin 1/17/22/23; Treg cell: regulatory T cell; TCR: T cell receptor; ILC: innate lymphoid cells; CD4+ T cell: cluster of differentiation 4 T cell; IgA: immunoglobulin A.

**Table 1 molecules-27-07622-t001:** Clinical study of common functional oligosaccharides in intestine-related diseases.

Functional Oligosaccharides	Study Title	Year	Conditions	Interventions	Actual Enrollment
FOS	Dietary Treatment of Crohn’s Disease	2006–2021	Crohn’s disease, inflammatory bowel disease	drug: active fructo-oligosaccharide drug: placebo fructo-oligosaccharide	73
scFOS	Effects of scFOS on Stool Frequency in People With Functionnal Constipation	2013–2018	functional constipation	dietary supplement: short-chain fructo-oligosaccharides dietary supplement: maltodextrin	120
IMO	Prebiotic Effects of Isomalto-oligosaccharide	2015–2017	intestinal microbiota,	dietary supplement: isomalto-oligosaccharide	54
GOS	GOS to Reduce Symptom Severity in IBS (EGIS)	2021–	irritable bowel syndrome, irritable bowel syndrome—constipation, irritable bowel syndrome—diarrhoea, irritable bowel syndrome—mixed	dietary supplement: galactooligosaccharides (GOS) dietary supplement: maltodextrine	210
HMO	Human Milk Oligosaccharides (HMOs) for Irritable Bowel Syndrome (IBS) (HIBS)	2022–	irritable bowel syndrome, IBS—irritable bowel syndrome	dietary supplement: human milk oligosaccharide mix other: placebo	500

**Table 2 molecules-27-07622-t002:** Sources of common functional oligosaccharides and their mode of action with probiotics.

Functional Oligosaccharide	Source	Composition	Advantage Probiotics	Transport Pathway	References
GOS	human milk, cow’s milk	monosaccharide and number: glucose 1, galactose 2–5; connection mode: β-1,4, β-1,6	*B. adolescentis*, *B. bifidum*, *B. longum*, *B. infantis*, *B. breve*, *B. animalis*, *B. catenulatum*; *L. reuteri*, *L. plantarum*, *L. paracasei*, *L. agili*, *L. fermentium*, *L. acidophilus*, *L. salivarius*, *L. casei*, *L. rhamnosus*, *L. bulgaricus*, *L. delbrueckii*, *Lactobacillus johnsonii*, *Lactobacillus gasseri*; *S. thermophilus*	LacEF, LacA, LacS, ABC, GPH, LacL, LacM	[36,42,43,44,45,46]
FOS	fruits, vegetables, honey, Jerusalem artichoke, cicory	monosaccharide and number: glucose 1, fructose 2–4; connection mode: α-1,2, β-1,2	*B. adolescentis*, *B. longum*, *B. breve*, *B. animalis*, *B. infantis*, *B. pseudolongum*; *L. reuteri*, *L. acidophilus*, *L. salivarius*, *L. plantarum*, *L. fermentium*, *L. casei*, *L. bulgaricus*; *Clostridium*, *Streptococcus*, *Coprococcus*, *Enterococcus*	PTS, ABC, MFS, LacS	[43,44,45,46,47,48,49]
IMO	corn steep liquor, honey, sugar cane juice	monosaccharide and number: glucose 2–5; connection mode: at least 1 α-1,6	*B. animalis*, *B. adolescentis*, *B. bifidum*, *B. longum*, *B. infantis*, *B. breve*; *L. plantarum*, *L. rhamnosus*, *L. paracasei*, *L. agilis*, *L. acidophilus*, *L. reuteri*, *L. lactic*, *L. delbrueckii*, *L. casei*; *S. lactic*, *S. thermophilus*	ABC, MalEFG-MsmK, PTS, MFS, MIP	[50,51,52]
XOS	birch, corncob, straw, bamboo	monosaccharide and number: xylose 2–7; connection mode: β-1,4	*B. adolescentis*, *B. longum*, *B. breve*, *B. animalis*, *B. catenulatum*, *B. pseudocatenulatum*, *B. thermophilum*; *L. plantarum*, *L. brevis*, *L. rhamnosus*, *L. fermentium*, *L. acidophilus*, *L. salivarius*, *L. casei*, *L. crispatus*, *L. lactis*, *L. mucosae*, *L. sakei*, *L. zeae*, *L. reuteri*; *Enterococcus faecalis and Enterococcus faecium*	ABC, MFS	[53,54,55]
COS	shrimp and crab shell, fungal cell wall	monosaccharide and number: N-acetyl-D-glucosamine 2–20; connection mode: β-1,4	*B. bifidium*; *L. brevis*, *L. casei*, *L. acidophilus*; *Akkermansia*, *S. thermophilus*	CsnEFG, SBP, PTS, ABC	[34,56,57,58,59]
HMO	breast milk, amniotic fluid	monosaccharide and number: glucose, N-acetyl-D-glucosamine, galactose, fucose, N-acetylneuraminic acid; connection mode: α-1,2, α-1,3, α-1,4, α-2,3, α-2,6	*B. infantis*, *B. longum*, *B. breve*, *B. bifidum*; *L. acidophilus*; *Bacteroides fragilis*, *Bacteroides vulgatus*, *Bacteroides thetaiotaomicron*	ABC	[60,61,62,63]

**Table 3 molecules-27-07622-t003:** The mode of action of common functional oligosaccharides on the intestinal immune barrier.

Functional Oligosaccharides	Immune Cells	Immune Factors	References
GOS	NK cells, T cells, phagocytes	increase IgA, IgM, IL-8, IL-10, IFN-γ; decrease IL-6, IL-18, IL-13, IL-33	[71]
FOS	B cells, T cells, macrophages, leukocytes	increase IgG, IgE, IFN-γ, IL-10; decrease IL-5, IL-6	[72,73,74,75]
IMO	T cells, phagocytes	increase lysozyme, IgE, IgG, IgA, IgM, IL-2, IFN-γ; decrease IL-5, IL-6, IL-13	[76,77,78]
XOS	B cells, T cells, NK cells, macrophages	increase IgG, IgA, IgM; decrease TLR2	[79]
COS	macrophages	increase CCL20, IgA, MHCII, TGF-131, pIgR; decrease CCL15, CCL25, ICL25, IL-1β, IL-4, IL-6, IL-8, IL-13, TNF-α	[80,81,82,83]
HMO	macrophages, T cell	increase INF-γ, IL-10 decrease IL-4, IL-6, IL-8, TNF-α, IL-1β, GM-CSF2, IL-17C, PF4, CXCL1, CCL20	[84,85,86,87,88,89]

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
