# Peer review of "Recent Research and Application Prospect of Functional Oligosaccharides on Intestinal Disease Treatment"

_molecules, 2022, doi:10.3390/molecules27217622_

Round 1
Reviewer 1 Report
Functional oligosaccharides are carbohydrates with a low degree of polymerization and exhibit beneficial effects on human intestinal health. Laboratory experiments and clinical studies indicated that functional oligosaccharides repair damaged intestinal tract and maintain intestinal homeostasis by regulating intestinal barrier function, immune response, and intestinal microbial composition.
The authors present herein an overview of the recent research progress on functional oligosaccharides on intestinal health. A a reader, I would like to know the stuctures of these functional oligosaccarides.
Thus, I suggest the authors should give one more figure to show these functional oligosaccarides (FOS, scFOS, IMO, GOS, HMO, ...... etc.).
In addition, it looks that there are many mistakes in reference part: ref 6, 9, 10, 12, 23, 31, 34, 36, 39, 90, 91, 93, 96, 100, ......,no pages; ref 13, 16, 33, 35, 42, 43, 44, 46, 50, 52, 54, 55-62, 70-77, 79, 80, 85, 86, 88, 101.......,no journal's name.
The authors should check the whole reference part carefully.
Author Response
Dear Reviewer,
Thank you for your decision and constructive comments on my manuscript. The 'revision' part shown in word that has been revised according to your comments. Revision notes, point-to-point, are given as follows:
According to your review opinions, we drew a schematic diagram of the structure of functional oligosaccharides, as shown in Figure 2. We also carefully examined and modified the format of all references.
Reviewer 2 Report
In this review article, the authors have presented an overview of the recent research progress on functional oligosaccharides on intestinal health. The manuscript is well organized by analysis of the related references. And a prospect of functional oligosaccharides in intestinal tract has been noted at the end of the paper. Thus, publication in Molecules is recommended after minor revisions
1) Recent advances related to “oligosaccharide” and “intestinal health” are recommended to be cited: Molecules 2022, 27(7), 2235; Molecules 2022, 27(19), 6672; Molecules 2022, 27(18), 5947; Molecules 2021, 26(8), 2199; Molecules 2021, 26(4), 1177.
2) Some of the information is missing for few of the references: 35, 54, 62, 76, 88, 111, 112, 122, 130, 132, etc.
Author Response
Dear Reviewer,
Thank you for your decision and constructive comments on my manuscript. The 'revision' part shown in word that has been revised according to your comments. Revision notes, point-to-point, are given as follows:
Based on your comments we have carefully read your recommended references and cited them as required. We also carefully examined and modified the format of all references.
Round 2
Reviewer 1 Report
Although the authors have revised the reference part, there are still many formating errors in this part. For example, ref 55 and 60 "xx xx = yy yy", only the journal names in ref 16, 36, 43, 62, 63, 65, 66, 68, 70, 72, 73, 82, 91, 96, 99, 109, 110, 111, 115, 117, 119, 121, 124, 125, 128, 131, 133, 134 and 136 are in abbreviated form. The authors should first figure out the citation format of this journal and then carefully check the reference part.
Author Response
Dear Reviewer,
Thank you for your decision and constructive comments on my manuscript. I apologize for the format errors of the references in our manuscripts. We have spent a long time on the manuscripts. Repeated additions and deletions of references have led to inconsistent formats. We have carefully considered the suggestion of Reviewer and make some changes.
According to your review, we carefully examined and modified the format of the manuscript 's references, but because some references do not have abbreviations, such as 5,15,55,113 and 126, we did not revise the format of these references.